# Migration through Resolution Cell Correction and Sparse Aperture ISAR Imaging for Maneuvering Target Based on Whale Optimization Algorithm—Fast Iterative Shrinkage Thresholding Algorithm

**DOI:** 10.3390/s24072148

**Published:** 2024-03-27

**Authors:** Xinrong Guo, Fengkai Liu, Darong Huang

**Affiliations:** 1Science College, Armed Police Engineering University, Xi’an 710051, China; rapgxr@163.com; 2Air and Missile Defense College, Air Force Engineering University, Xi’an 710051, China; liufengkai173@163.com

**Keywords:** inverse synthetic aperture radar (ISAR), maneuvering target, migration through resolution cell (MTRC), whale optimization algorithm (WOA)

## Abstract

Targets faced by inverse synthetic aperture radar (ISAR) are often non-cooperative, with target maneuvering being the main manifestation of this non-cooperation. Maneuvers cause ISAR imaging results to be severely defocused, which can create huge difficulties in target identification. In addition, as the ISAR bandwidth continues to increase, the impact of migration through resolution cells (MTRC) on imaging results becomes more significant. Target non-cooperation may also result in sparse aperture, leading to the failure of traditional ISAR imaging algorithms. Therefore, this paper proposes an algorithm to realize MTRC correction and sparse aperture ISAR imaging for maneuvering targets simultaneously named whale optimization algorithm–fast iterative shrinkage thresholding algorithm (WOA-FISTA). In this algorithm, FISTA is used to perform MTRC correction and sparse aperture ISAR imaging efficiently and WOA is adopted to estimate the rotational parameter to eliminate the effects of maneuvering on imaging results. Experimental results based on simulation and measured datasets prove that the proposed algorithm implements sparse aperture ISAR imaging and MTRC correction for maneuvering targets simultaneously. The proposed algorithm achieves better results than traditional algorithms under different signal-to-noise ratio conditions.

## 1. Introduction

The biggest challenge to inverse synthetic aperture radar (ISAR) imaging is the non-cooperative nature of the targets. Due to the time accumulation required for ISAR imaging, the motion of the target during ISAR imaging can greatly affect the quality of ISAR imaging. Based on the spatial relationship between the direction of motion and the direction of the radar line of sight, the motion of the target can be decomposed into translational and rotational motion. The translational motion is parallel to the radar line-of-sight direction, which does not benefit ISAR imaging and needs to be compensated for [1,2,3,4]. The rotational motion contains the azimuth information of the target, which is needed for ISAR imaging. The classical range–Doppler (RD) algorithm treats rotational motion as a uniform velocity, which holds when the target is flying smoothly. However, when the target is maneuvering, the rotational motion is non-uniform and RD algorithm will be invalid. Therefore, it is essential to conduct research on ISAR imaging of maneuvering targets.

Many researchers have already contributed to the study of ISAR imaging of maneuvering targets. Most of them believe that the non-uniform rotation of a maneuvering target can be viewed as uniformly accelerated. Therefore, the echo signal of a maneuvering target is a multi-component linear frequency modulation (m-LFM) signal. Based on this signal model, two kinds of ISAR imaging algorithms for maneuvering targets with different principles are proposed. Some researchers have obtained well-focused ISAR images of maneuvering targets by acquiring the instantaneous Doppler frequency of the echoes through time-frequency transformations. Therefore, this kind of algorithm is called the range-instantaneous Doppler (RID) method. Typical RID methods include Wigner–Ville distribution [5,6,7] and Radon–Wigner transform [8,9]. In addition, some researchers combine RID methods with compressed sensing techniques for sparse aperture ISAR imaging of maneuvering targets [10,11]. RID methods are easy to implement and have high computational efficiency. However, the resolution of the RID method is not satisfactory and time-frequency transformations tend to introduce cross-terms and lead to false peaks.

Other researchers reconstruct the echo by estimating the frequency and chirp rate of the m-LFM signal. This kind of algorithm is called the parameter estimation (PE) method. Common PE methods include LV’s distribution (LVD) [12,13,14], chirp–Fourier transform [15,16,17], and high-order ambiguity function (HAF) [18,19]. Compared with the RID method, the PE method can effectively suppress cross terms and obtain imaging results with higher resolution. However, PE methods have low computational efficiency. Additionally, both RID and PE methods are invalid for the echoes with sparse aperture.

In rotational motion, if the scattering center varies by more than one range cell in the radar line-of-sight direction, this results in migration through resolution cells (MTRC). Similar to the maneuvering of the targets, MTRC also causes defocused ISAR images. Currently, the most commonly used MTRC correction method is the Keystone [20]. More efficient and better-integrated MTRC correction methods need to be proposed urgently.

Non-cooperation of the target may also lead to unavailability of part of the echo pulse, i.e., sparse aperture. Most full-aperture ISAR imaging algorithms are unable to handle sparse aperture echoes due to uneven pulse sampling. Compressed sensing techniques are capable of recovering sparse signals and are therefore an important tool for addressing sparse aperture ISAR imaging. Some classical compressed sensing algorithms such as OMP [21], SL0 [22], and sparse Bayes [23] have been used to implement sparse aperture ISAR imaging. However, most sparse aperture ISAR imaging algorithms do not attempt to implement MTRC correction.

In sum, this paper proposes an algorithm to realize MTRC correction and ISAR imaging for maneuvering targets simultaneously named whale optimization algorithm–fast iterative shrinkage thresholding algorithm (WOA-FISTA). Specifically, we first perform a 2D sparse representation of the echoes of the maneuvering target. FISTA is used to perform MTRC correction and ISAR imaging efficiently and WOA is adopted to estimate the rotational parameter to eliminate the effects of maneuvering on imaging results. Alternating iterations of FISTA and WOA resulted in well-focused maneuvering target imaging results. Experimental results based on simulation and measured datasets prove that the proposed algorithm is well-performed for sparse aperture echoes and robust for noise.

## 2. Signal Model

Most ISAR systems transmit broadband LFM signals. The expression of the LFM signal can be written as
(1)$$s\left(\hat{t},t_{m}\right)=\mathrm{rect}\left(\frac{\hat{t}}{T_{p}}\right)\exp\left[j2\pi f_{c}t\right]\cdot \exp\left[j\pi \mu {\hat{t}}^{2}\right]$$
where $T_{p}$ is the pulse width of the transmitted signal, $\mathrm{rect}\left(\;\right)$ is the rectangular window function, $f_{c}$ is the carrier frequency, $t=\hat{t}+t_{m}$ is the full time, $\hat{t}$ is a fast time, $t_{m}$ is a slow time, and μ is the chirp rate.

The transmitted signal is reflected by the target and then received by the ISAR system, which can be expressed as
(2)$$\begin{array}{cc} s_{R}(\hat{t},t_{m})= & \sum_{p=1}^{N}\sigma_{p}\mathrm{rect}\left(\frac{\hat{t}-\frac{2R_{p}\left(t_{m}\right)}{c}}{T_{p}}\right)\exp\left[j2\pi f_{c}\left(t-\frac{2R_{p}\left(t_{m}\right)}{c}\right)\right]\cdot \\ & \exp\left[j\pi \mu {\left(\hat{t}-\frac{2R_{p}\left(t_{m}\right)}{c}\right)}^{2}\right] \end{array}$$
where *p* is the index of scattering centers and $R_{p}\left(t_{m}\right)=R_{tran}\left(t_{m}\right)+R_{rot,p}\left(t_{m}\right)$ is the slant range. $R_{tran}\left(t_{m}\right)$ is the translational motion. After matched filtering and translational motion compensation, the echo can be rewritten as
(3)$$s_{R}(\hat{t},t_{m})=\sum_{p=1}^{N}\sigma_{p}\exp\left[-j\frac{4\pi }{c}\left(f_{c}+f_{r}\right)R_{rot,p}\left(t_{m}\right)\right]$$
where $f_{r}=\mu \hat{t}$ is the corrected fast time-frequency.

The turntable model of ISAR imaging is shown in Figure 1.

According to Figure 1, $R_{rot,p}\left(t_{m}\right)$ can be written as
(4)$$R_{rot,p}\left(t_{m}\right)=x_{p}\sin\theta \left(t_{m}\right)+y_{p}\cos\theta \left(t_{m}\right)$$
where $y_{p}$ and $x_{p}$ are the Y–X coordinates of scattering center *p* and $\theta \left(t_{m}\right)$ is the rotational angle. Currently, most scholars believe that maneuvering targets undergo uniform acceleration rotation during ISAR imaging. Thus, $\theta \left(t_{m}\right)$ can be written as
(5)$$\theta \left(t_{m}\right)=\omega t_{m}+\frac{1}{2}a{t_{m}}^{2}$$
where ω is the rotational angular velocity and a is the rotational angular acceleration. Considering that the rotation angle required for ISAR imaging is 3–5°, (3) can be approximated as
(6)$$R_{rot,p}\left(t_{m}\right)=x_{p}\omega t_{m}+\frac{1}{2}x_{p}a{t_{m}}^{2}+y_{p}$$

Substituting (6) into (3), the echo can be written as
(7)$$s_{R}(\hat{t},t_{m})=\sum_{p=1}^{N}\sigma_{p}\exp\left[-j\frac{4\pi }{c}\left(f_{c}+f_{r}\right)\left(x_{p}\omega t_{m}+\frac{1}{2}x_{p}a{t_{m}}^{2}+y_{p}\right)\right]$$
where the target maneuver caused a slow time quadratic phase, and the coupling of fast time to slow time caused the MTRC.

Sampling $f_{r}$, $t_{m}$, x, and y, the 2D sparse representation of the echo is expressed as
(8)$$S=\Phi D\left(\omega \right)X+n$$
(9)$$D\left(\eta \right)=\left(\begin{array}{cccccc} d\left(0,0,0,0\right) & \ldots & d\left(0,0,0,N-1\right) & d\left(0,0,1,0\right) & \ldots & d\left(0,0,M-1,N-1\right)\\ \vdots & \ddots & & \vdots & & \vdots \\ d\left(L-1,0,0,0\right) & & & & & \\ d\left(0,1,0,0\right) & & \ddots & \vdots & & \vdots \\ \vdots & & & & & \\ d\left(L-1,P-1,0,0\right) & & \ldots & \ldots & & d\left(L-1,P-1,M-1,N-1\right) \end{array}\right)$$
(10)$$\begin{array}{c} d\left(l,p,m,n\right)\\ =\exp\left[-j\frac{2\pi }{f_{s}T_{p}}nl\right]\cdot \\ \exp\left[-j\frac{4\pi }{c}\left(f_{c}+\frac{B}{f_{s}T_{p}}l\right)\frac{\lambda }{2P}mp\right]\cdot \\ \exp\left[-j\frac{\pi }{c}\left(f_{c}+\frac{B}{f_{s}T_{p}}l\right)\frac{\eta \lambda mp^{2}}{P\cdot prf}\right] \end{array}$$
(11)$$X=\left(\begin{array}{c} \sigma \left(0,0\right)\\ \vdots \\ \begin{array}{c} \sigma \left(0,N-1\right)\\ \sigma \left(1,0\right) \end{array}\\ \vdots \\ \sigma \left(M-1,N-1\right) \end{array}\right)$$
(12)$$s_{R}\left(l,p\right)=\left\{\begin{array}{c} \begin{array}{cc} s_{R}\left(l,p\right) & p\notin \Phi_{n} \end{array}\\ \begin{array}{cc} 0 & p\in \Phi_{n} \end{array} \end{array}\right.$$
where S is the echo, X is the scattering center amplitude, n is the noise, $\eta =\frac{a}{\omega }$ is the rotational parameter, Φ is the measured matrix, it is formed by 0 and 1 and decided by (12), $D\left(\eta \right)$ is the dictionary matrix, $\Phi_{n}$ is invalid apertures, n is the noise, ⊙ is Hadamard product and m, n, l, p are the index of azimuth coordinate, range coordinate, range cell, cross-range cell, respectively. The maneuvering target imaging can be completed by solving X from (8).

According to the compressed sensing theory, a focused ISAR image of a maneuvering target can be obtained by solving the following optimization problem
(13)$$\hat{X}=\arg\min{\left\|X\right\|}_{0}\;\;s.t.\;S=\Phi D\left(\omega \right)X$$
where ${\left\|\;\right\|}_{0}$ is the L0-norm. Considering the L0-norm causes NP-hard problems, we replace L0- norm by L1-norm and (13) can be modified as
(14)$$\hat{X}=\arg\min\left\{\frac{1}{2}{\left\|S-\Phi D\left(\omega \right)X\right\|}_{2}^{2}+\delta {\left\|X\right\|}_{1}\right\}$$
where δ is used to control the sparse degree of the solution.

## 3. The Principle of WOA-FISTA

This section describes the principles and detailed steps of WOA-FISTA. According to the principle of proximal regularization [24,25], (14) can be expressed as
(15)$$X^{i+1}=\underset{X}{\arg\min}\left\{\frac{1}{2}{\left\|X-\left(X^{i}+\nabla f\left(X^{i}\right)\right)\right\|}_{2}^{2}+\delta {\left\|X\right\|}_{1}\right\}$$
where $f\left(X\right)=\frac{1}{2}{\left\|S-\Phi D\left(\omega \right)X\right\|}_{2}^{2}$, *i* is the number of iterations, and ∇ is the gradient operator. We are able to calculate $\nabla f\left(X\right)$ as
(16)$$\nabla f\left(X\right)={\left(\Phi D\left(\omega \right)\right)}^{H}\left(S-\Phi D\left(\omega \right)X\right)$$
where *H* denotes the conjugate transpose of the matrix.

According to (15), any element of X satisfies
(17)$$x^{i+1}=\underset{x}{\arg\min}\left\{\frac{1}{2}{\left|x-z\right|}^{2}+\delta \left|x\right|\right\}$$
where $z=x^{i}+\nabla f\left(x^{i}\right)$. Based on the trigonometric inequality, we can obtain
(18)$$\left|x-z\right|\geq \left|x\right|-\left|z\right|$$

Therefore, (17) is equivalent to the following equation
(19)$$x^{i+1}=\underset{x}{\arg\min}\left\{\frac{1}{2}{\left(\left|x\right|-\left|z\right|\right)}^{2}+\delta \left|x\right|\right\}$$

Solving the optimization problem in (19), we are able to obtain that
(20)$$\left|x^{i+1}\right|=\left\{\begin{array}{l} \left(\left|z\right|-\zeta \right),\left|z\right|\geq \zeta \\ 0,\left|z\right|<\zeta \end{array}\right.$$

Considering that (18) takes the equality sign conditional on $\frac{x}{\left|x\right|}=\frac{z}{\left|z\right|}$, the x corresponding to the minimum value in (17) can be expressed as
(21)$$x^{i+1}=\mathrm{prox}(z)=\left\{\begin{array}{l} \frac{z}{\left|z\right|}\left(\left|z\right|-\zeta \right),\left|z\right|\geq \zeta \\ 0,\left|z\right|<\zeta \end{array}\right.$$

Since z is only related to $x^{i}$, we can use the iterative process in (21) to implement MTRC correction and ISAR imaging. However, we need to correct z to improve the iterative efficiency of the algorithm. Nesterov acceleration can increase the convergence speed of the algorithm with very little additional computation [26]. After adopting the Nesterov acceleration, the corrected z can be written as
(22)$$z=x^{i}+\nabla f\left(x^{i}\right)+\frac{i-2}{i+1}\left(x^{i}-x^{i-1}\right)$$
where *i* is the number of iterations.

Based on the above principle, the pseudocode of FISTA is shown in Algorithm 1.
**Algorithm 1.** The pseudocode of FISTA**Input:** original echo S, measured matrix Φ, dictionary matrix $D\left(\eta \right)$, iteration number *I*, shrinkage threshold ζ

**Output:** Imaging result without MTRC $\hat{X}$
 1. Initialize $X^{1}={\left(\Phi D\left(\omega \right)\right)}^{T}S$
2. **for** i = 1 **to** *I* do3. Calculate $\nabla f\left(X^{i}\right)$
4. Update $z^{i}$ as 
$z^{i}=X^{i}+\nabla f\left(X^{i}\right)+\frac{i-2}{i+1}\left(X^{i}-X^{i-1}\right)$
 5. Update $X^{i}$ as 
$X^{i+1}=\mathrm{prox}(z)$
6. **end**7. **Return**
$\hat{X}=X^{I+1}$


It is not difficult to notice that (14) contains an unknown parameter, i.e., the rotational parameter η. We must accurately estimate η, otherwise FISTA will not give the correct imaging results. We adopt the principle of minimum mean square error to estimate η, which can be expressed as
(23)$$\hat{\eta }=\arg\min\left\{{\left\|S-\Phi D\left(\omega \right)X\right\|}_{2}^{2}\right\}$$

Since the gradient descent algorithm is difficult to solve the problem in (23), we introduce the WOA algorithm. WOA was inspired by the predatory behavior of whales. It does not need to solve for the derivatives of the objective function and is not dependent on the initial solution [27,28,29].

WOA has two key parameters ρ and $Z=2\vartheta \cdot rand_{1}+\vartheta$ that control the algorithm operation. ρ and $rand_{1}$ are random variables between 0 and 1. During the entire iteration ϑ decreases linearly from 2 to 0. When ρ≥0.5, each individual spirals toward the best individual. This process can be represented as
(24)$$\xi_{i}^{t+1}=\left|\xi_{best}^{t}-\xi_{i}^{t}\right|\exp\left[dw\right]\cos\left(2\pi w\right)+\xi_{best}^{t}$$
where $\xi_{best}^{t}$ is the optimal individual in the *t*th generation, w is a random variable between −1 and 1, and d is the spiral parameter. d is used to control the spiral movement in WOA.

If ρ<0.5, the value of Z will determine the iteration strategy. When $\left|Z\right|<1$, all individuals move towards the optimal individual. When $\left|Z\right|\geq 1$, each individual moves randomly to other individuals. Under this condition, the process of updating can be expressed as
(25)$$\xi_{i}^{t+1}=\left\{\begin{array}{l} \xi_{best}^{t}-Z\left|rand_{2}\cdot \xi_{best}^{t}-\xi_{i}^{t}\right|,\left|Z\right|<1\\ \xi_{rand}^{t}-Z\left|rand_{2}\cdot \xi_{rand}^{t}-\xi_{i}^{t}\right|,\left|Z\right|\geq 1 \end{array}\right.$$
where $\xi_{rand}^{t}$ is a randomly selected individual in the *t*th generation and $rand_{2}$ is a random variable between 0 and 2. It can be seen that WOA not only utilizes the information of the optimal individual but also makes full use of the information of other individuals within the population. Therefore, compared with PSO [30], WOA makes it easier to find the global optimal solution.

The pseudocode of WOA is shown in Algorithm 2.
**Algorithm 2.** The pseudocode of WOA**Input:** Maximum number of iterations max, the spiral parameter d, number of individuals num

**Output:**
$\hat{\eta }$
1. Generate initial population randomly  2. **Repeat**
 3. Calculate ρ and Z
 4. **If** ρ≥0.5 **then**
 5. Update the location of all individuals as (24)  6. **Else**
 7. Update the location of all individuals as (25)  8. **End if**
 9. **Until** t=max
 10. **return** $\hat{\eta }=\eta_{gbest}^{Q}$


In sum, the flowchart of WOA-FISTA is shown in Figure 2.

WOA and FISTA were alternately iterated to obtain more accurate ISAR imaging results. We also employ asynchronous iterations to ensure convergence of WOA and FISTA. Specifically, each time the loop shown in Figure 2 is executed, FISTA is updated once and WOA is updated Q times.

## 4. Experiments

In this section, we use simulated and measured datasets to validate the effectiveness of the proposed method. The simulated aircraft contains 330 scattering centers. We set the radar carrier frequency at 10 GHz, bandwidth at 2 GHz, number of pulses at 256, and pulse repetition frequency (PRF) at 200 Hz. The origin of the target coordinate system is 20 km away from the radar. The angular velocity is 0.03 rad/s and the angular acceleration is 0.01 rad/s^2^. The shape of the simulated aircraft is shown in Figure 3.

The parameters of WOA-FISTA are set to I = 4, L = 5, Q = 10, d = 1. We adopt the RD algorithm, chirp–Fourier transform method [16], and sparse reconstruction-range-instantaneous Doppler (SR-RID) [10] as control groups, where RD algorithm is the most common ISAR imaging algorithms, chirp–Fourier transform method is a typical PE method and SR-RID is an algorithm that combines RID methods with compressed sensing techniques. Under the conditions of signal-to-noise ratio (SNR) is 10 dB, the imaging results of the above four methods are shown in Figure 4. Entropy and contrast are used to quantify imaging quality. The definition of entropy and contrast can be written as
(26)$$EN=\sum_{b=1}^{B}\sum_{a=1}^{A}\frac{{\left|IMG\left(a,b\right)\right|}^{2}}{{\left\|IMG\right\|}_{l2}^{2}}\ln\frac{{\left\|IMG\right\|}_{l2}^{2}}{{\left|IMG\left(a,b\right)\right|}^{2}}$$
(27)$$IC=\frac{\sqrt{\sum_{b=1}^{B}\sum_{a=1}^{A}\left(\left|IMG\left(a,b\right)\right|-\frac{{\left\|IMG\right\|}_{l1}}{AB}\right)}}{{\left\|IMG\right\|}_{l1}}$$
where IMG is the ISAR image, A and B, respectively, represent the total number of rows and columns of pixel. We provide the entropy and contrast corresponding to the imaging results in Table 1.

Then we conducted another experiment in a low SNR environment to verify the noise immunity of WOA-FISTA. We still compare WOA-FISTA with RD, chirp–Fourier, and SR-RID. Set SNR = −5 dB, the imaging result of the above four methods are shown in Figure 5. We also provide the entropy and contrast corresponding to the imaging results in Table 2.

Next, we verify the performance of WOA-FISTA for sparse aperture echoes. Set SNR = 10 dB and the pulse sampling rate is 50%, the imaging result of the above four methods are shown in Figure 6. Similarly, we provide the entropy and contrast corresponding to the imaging results in Table 3.

Next, we adopt maneuvering Yak-42 measured dataset to validate the effectiveness of WOA-FISTA. A C-band radar records the dataset with a bandwidth of 400 MHz, a PRF of 100 Hz, and a carrier frequency of 5.52 GHz. The dataset contains 256 pulses; each pulse has 256 sampling points. We provide the imaging result of RD, chirp–Fourier, SR-RID, and WOA-FISTA in Figure 7 and the entropy and contrast of the above methods in Table 4.

Then we validate the noise robustness of WOA-FISTA using the measured Yak-42 dataset. Set SNR = 0 dB, the imaging result of the above four methods, is shown in Figure 8. We also provided the entropy and contrast of the imaging results in Table 5.

Next, we validate the performance of the above four methods for sparse aperture echo. The pulse sampling rate is 75%, 50%, and 25%, respectively. The imaging results of the above four methods are shown in Figure 9, Figure 10 and Figure 11, respectively. We provide the entropy and contrast of the imaging results in Table 6, Table 7 and Table 8.

## 5. Discussion

We can find in Figure 4 and Table 1 that the imaging result of RD is severely defocused, especially the wings. These defocused scattering centers span multiple range bins. The above phenomenon amply demonstrates the significant impact of target maneuvering and MTRC on ISAR imaging. Chirp–Fourier solves the defocusing to some extent but introduces false peaks. This makes the chirp–Fourier result less than ideal. The SR-RID results do not have false peaks, but the defocusing at the wings is not completely solved. Both chirp–Fourier and SR-RID failures are due to unsuccessful correction of the MTRC. In comparison, the result of WOA-FISTA is well-focused and has the lowest entropy and highest contrast. These experimental results demonstrate that WOA-FISTA can effectively eliminate the effects of target maneuvers and MTRC on ISAR imaging.

In Figure 5 and Table 2, the strong background noise further reduces the quality of RD results. Chirp–Fourier eliminates most of the noise but still retains a large number of false peaks. SR-RID has slightly less noise rejection than chirp–Fourier. In contrast, the results of WOA-FISTA are almost free of false peaks and have the lowest entropy and highest contrast. These results demonstrate the good noise rejection capability of WOA-FISTA.

According to Figure 6 and Table 3, the result of RD shows many false peaks caused by the sparse aperture. The result of chirp–Fourier is also disturbed by many false peaks. This is because chirp–Fourier does not apply to the echoes with sparse aperture. SR-RID achieves better results than chirp–Fourier, but still cannot eliminate the effect of sparse aperture. In contrast, WOA-FISTA eliminates the effect of sparse aperture. These results demonstrate that WOA-FISTA is effective for sparse aperture echoes.

It can be found from Figure 7 and Table 4 that the result of RD is significantly defocused, which proves that this dataset is from a maneuvering target. Chirp–Fourier eliminates the defocusing to some extent, but the scattering center of the nose is missing. The result of SR-RID also has some degree of defocusing. These are caused by the MTRC. In comparison, WOA-FISTA achieved the sharpest imaging results. These experiments based on the measured dataset further demonstrate the superior performance of WOA-FISTA.

Under the condition of SNR = 0 dB, the imaging results of RD showed a lot of noise. The imaging results of chirp–Fourier were similar to the full aperture experiments. This is because chirp–Fourier adopts the CLEAN technique, which can effectively suppress the noise. The imaging results of SR-RID showed a small amount of noise. However, the imaging results of WOA-FISTA remained well-focused. These results further prove that WOA-FISTA is noise-robust.

At a pulse sampling rate of 75%, a small number of false peaks appeared in the imaging results of RD. The imaging results of chirp–Fourier were in general agreement with the full-aperture experiments. This is because Yak-42 has fewer scattering centers and is less affected by the sparse aperture. The imaging results of SR-RID still show some degree of scattering. In contrast, the imaging results of WOA-FISTA have the highest quality.

At a pulse sampling rate of 50%, the RD imaging results showed more false peaks. The performance of the chirp–Fourier was significantly degraded. This is because the CLEAN technique does not apply to sparse aperture echoes. The imaging results of SR-RID showed no significant change. The performance of WOA-FISTA was not affected by the missing echoes.

At a pulse sampling rate of 25%, the RD and chirp–Fourier imaging results were heavily defocused. A few false peaks were also observed in the SR-RID imaging results. In contrast, the imaging results of WOA-FISTA are still clear and well-focused. These experiments fully demonstrate the applicability of WOA-FISTA to sparse aperture echoes.

## 6. Conclusions

This paper proposes a novel ISAR imaging algorithm to realize MTRC correction and ISAR imaging for maneuvering targets simultaneously named WOA-FISTA. Based on the 2D sparse representation of the echoes of the maneuvering target, FISTA is used to perform MTRC correction and ISAR imaging efficiently and WOA is adopted to estimate the rotational parameter to eliminate the effects of maneuvering on imaging results. Alternating iterations of FISTA and WOA resulted in well-focused maneuvering target imaging results. Experimental results based on simulation and measured datasets prove that the proposed algorithm is well-performed for sparse aperture echoes and robust for noise.

It is worth mentioning that the parameters of WOA-FISTA rely on empirical selection. The performance of WOA-FISTA can be significantly degraded if the parameters are not properly selected. Currently, the research of combining deep learning and compressed perception is of great interest. In future research, we will focus on deep learning-based ISAR imaging techniques that rely on neural networks for automatic parameter tuning.

## Figures and Tables

**Figure 1 sensors-24-02148-f001:**
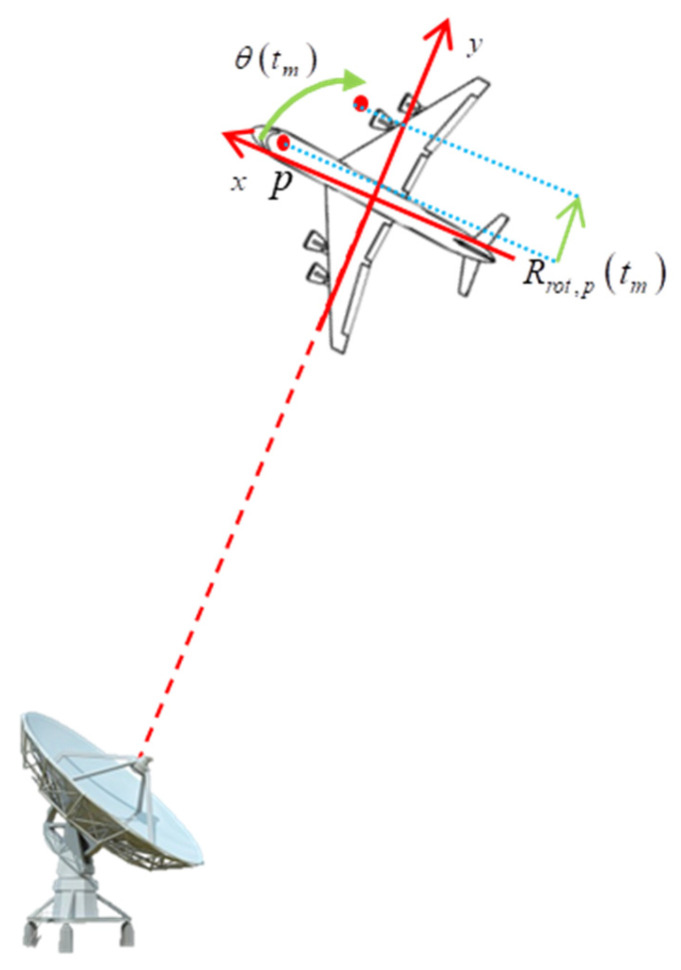
The turntable model of ISAR imaging.

**Figure 2 sensors-24-02148-f002:**
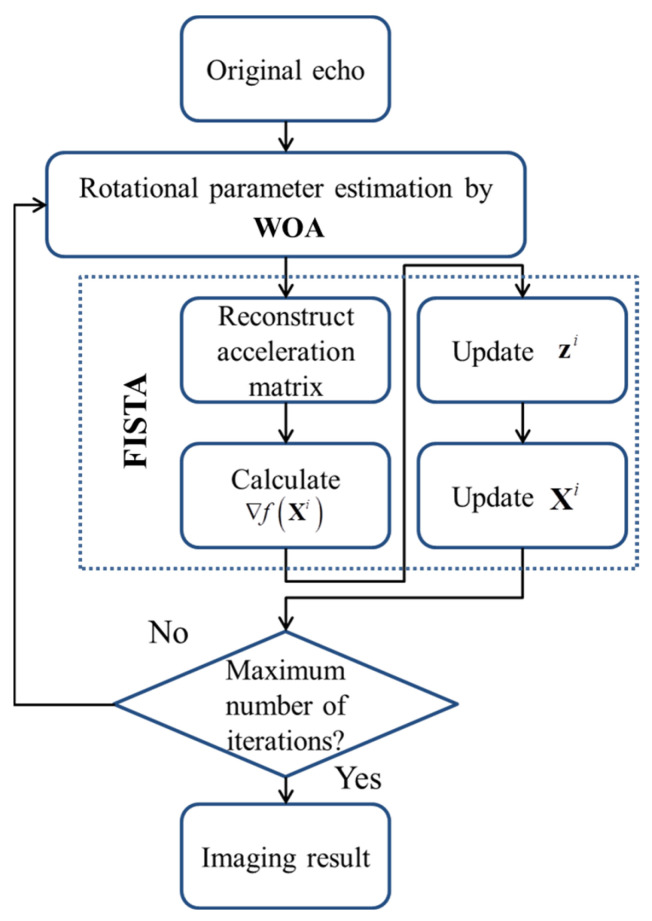
The flowchart of WOA-FISTA.

**Figure 3 sensors-24-02148-f003:**
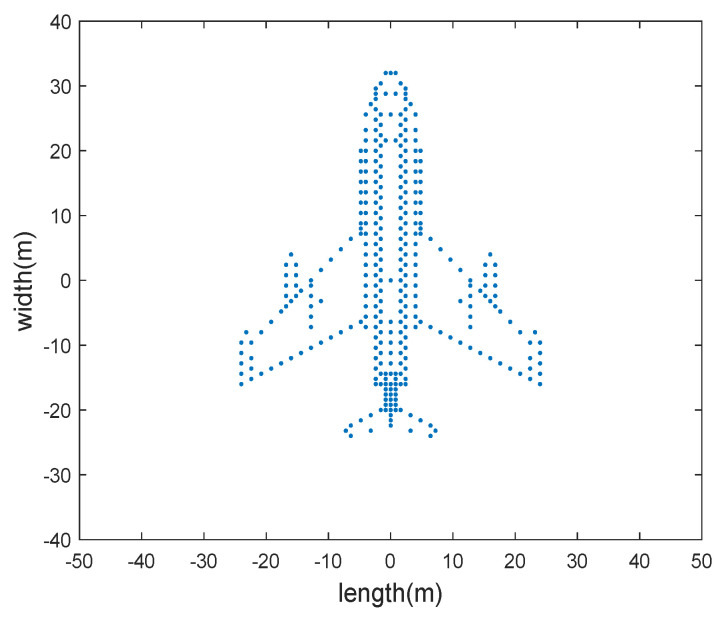
The shape of the simulated aircraft.

**Figure 4 sensors-24-02148-f004:**
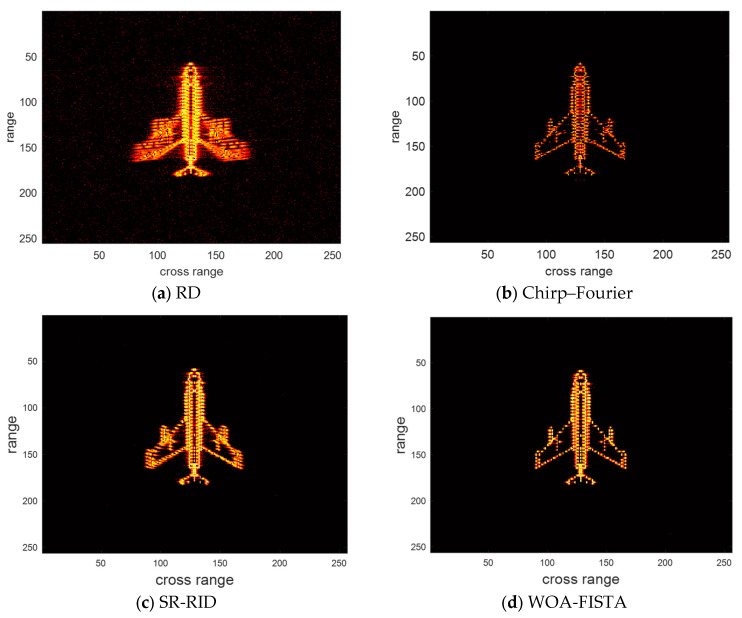
The imaging results of different methods under SNR = 10 dB.

**Figure 5 sensors-24-02148-f005:**
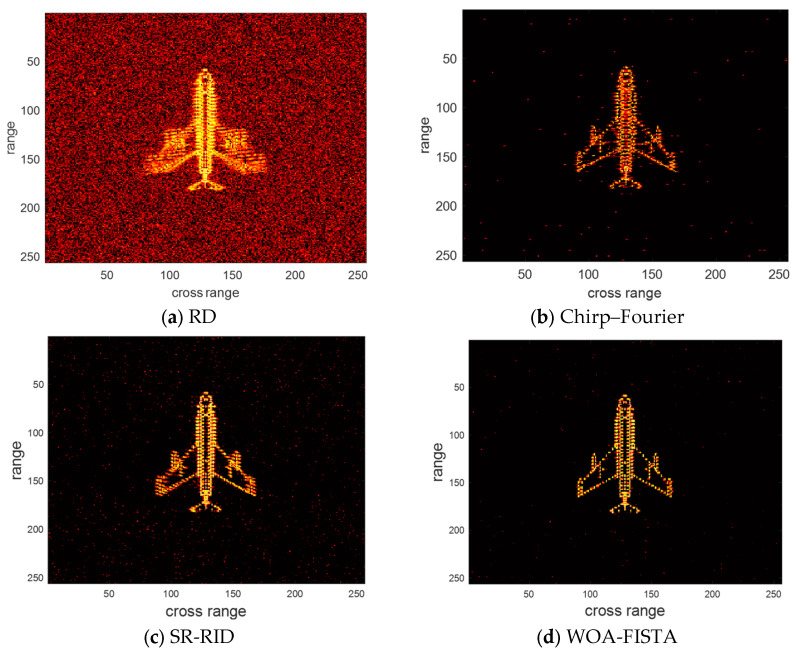
The imaging results of different methods under SNR = −5 dB.

**Figure 6 sensors-24-02148-f006:**
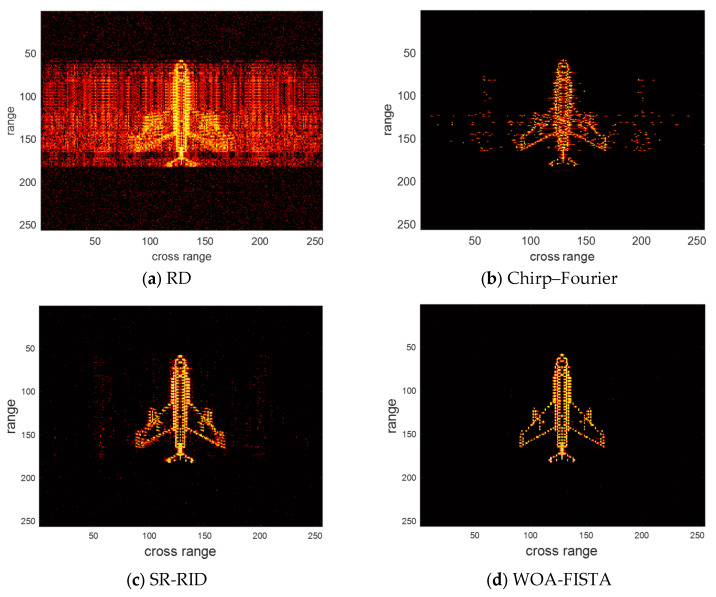
The imaging results of different methods under pulse sampling rate is 50%.

**Figure 7 sensors-24-02148-f007:**
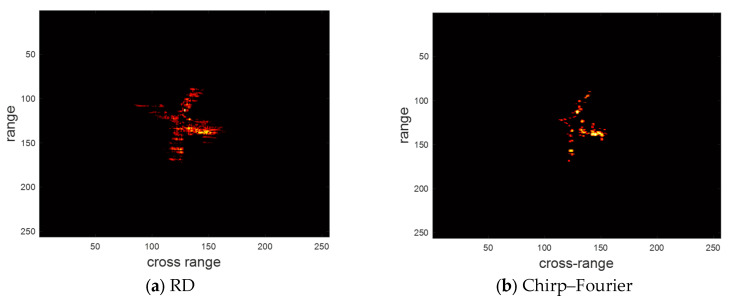
The imaging results of different methods for the measured dataset.

**Figure 8 sensors-24-02148-f008:**
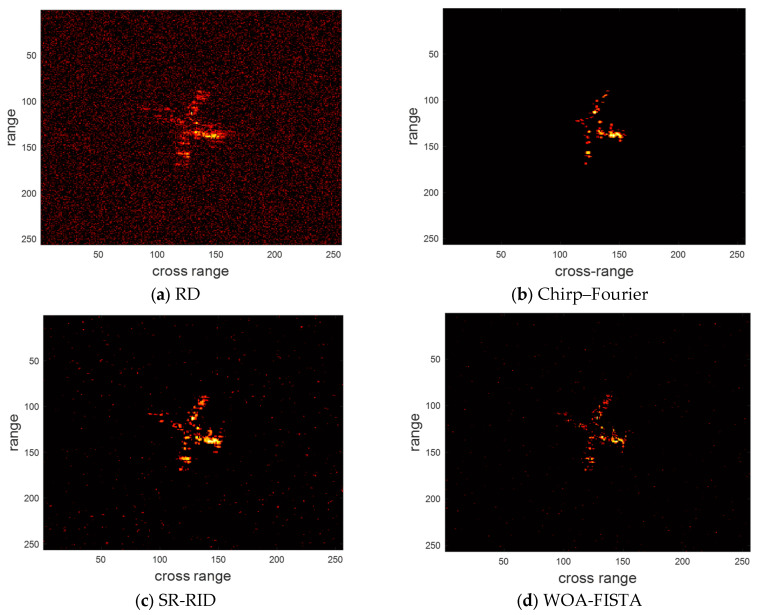
The imaging results of different methods for low SNR in the measured dataset.

**Figure 9 sensors-24-02148-f009:**
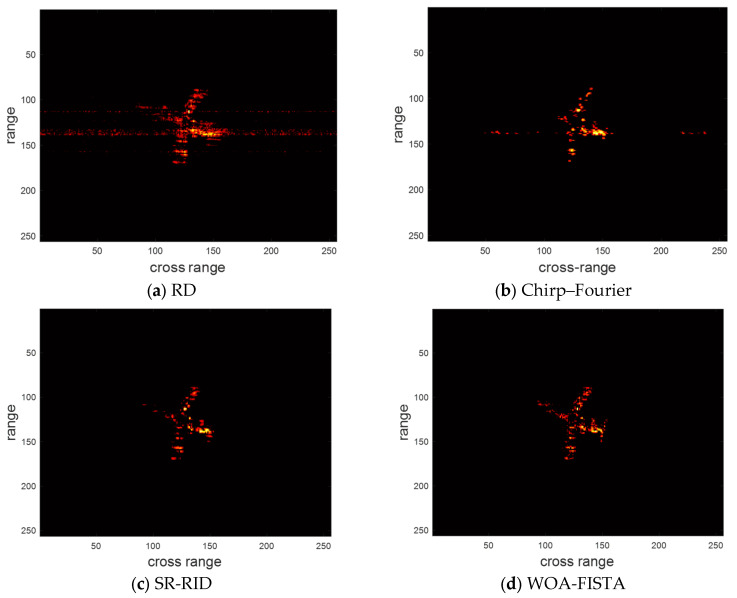
The imaging results of different methods for the measured dataset under pulse sampling rate of 75%.

**Figure 10 sensors-24-02148-f010:**
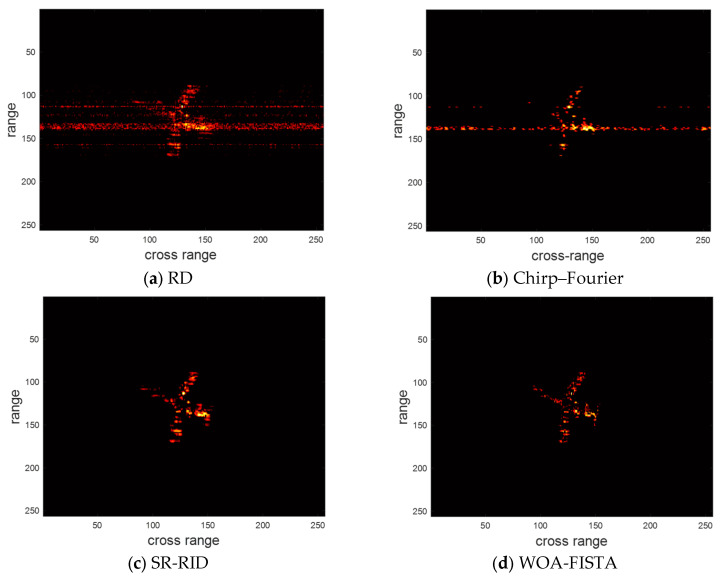
The imaging results of different methods for the measured dataset under pulse sampling rate of 50%.

**Figure 11 sensors-24-02148-f011:**
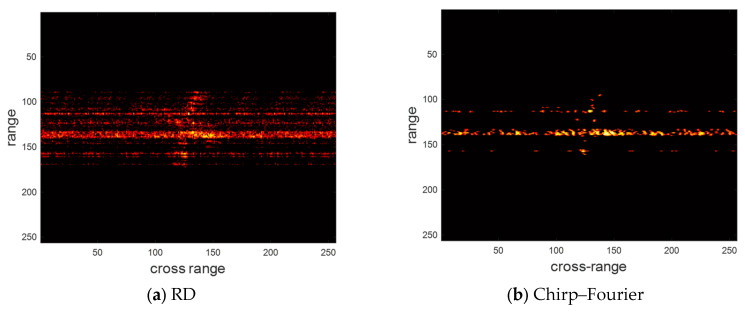
The imaging results of different methods for the measured dataset under pulse sampling rate of 25%.

**Table 1 sensors-24-02148-t001:** The entropy and contrast of different methods under SNR = 10 dB.

	RD	Chirp–Fourier	SR-RID	WOA-FISTA
Entropy	8.6532	7.2164	7.4761	6.7806
Contrast	5.3621	8.6965	9.2114	11.3673

**Table 2 sensors-24-02148-t002:** The entropy and contrast of different methods under SNR = −5 dB.

	RD	Chirp–Fourier	SR-RID	WOA-FISTA
Entropy	9.8706	7.9654	8.2361	7.4120
Contrast	4.2034	8.1012	7.3644	10.0287

**Table 3 sensors-24-02148-t003:** The entropy and contrast of different methods under pulse sampling rate is 50%.

	RD	Chirp–Fourier	SR-RID	WOA-FISTA
Entropy	8.8768	7.8175	7.3484	6.6332
Contrast	3.7621	6.7904	8.0337	10.9128

**Table 4 sensors-24-02148-t004:** The entropy and contrast of different methods for the measured dataset.

	RD	Chirp–Fourier	SR-RID	WOA-FISTA
Entropy	7.8936	7.0361	7.3685	6.7612
Contrast	6.1341	8.8696	8.1777	9.6532

**Table 5 sensors-24-02148-t005:** The entropy and contrast of different methods for the low SNR measured dataset.

	RD	Chirp–Fourier	SR-RID	WOA-FISTA
Entropy	8.6212	7.2736	7.8310	6.9367
Contrast	4.8775	8.3906	7.5161	9.0168

**Table 6 sensors-24-02148-t006:** The entropy and contrast of different methods for the measured dataset under pulse sampling rate of 75%.

	RD	Chirp–Fourier	SR-RID	WOA-FISTA
Entropy	8.1366	7.3736	7.5216	6.8102
Contrast	5.2047	8.6627	8.1690	9.3679

**Table 7 sensors-24-02148-t007:** The entropy and contrast of different methods for the measured dataset under pulse sampling rate of 50%.

	RD	Chirp–Fourier	SR-RID	WOA-FISTA
Entropy	8.3961	7.7360	7.5933	6.8416
Contrast	4.7648	7.6724	8.1593	9.0167

**Table 8 sensors-24-02148-t008:** The entropy and contrast of different methods for the measured dataset under pulse sampling rate of 25%.

	RD	Chirp–Fourier	SR-RID	WOA-FISTA
Entropy	8.8363	8.2135	7.6714	6.9726
Contrast	4.2165	6.3408	7.7109	8.8124

## Data Availability

The data presented in this study are available on request from the corresponding author. The data are not publicly available due to the request of the funder.

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
