# Peer review of "Migration through Resolution Cell Correction and Sparse Aperture ISAR Imaging for Maneuvering Target Based on Whale Optimization Algorithm—Fast Iterative Shrinkage Thresholding Algorithm"

_sensors, 2024, doi:10.3390/s24072148_

Round 1

Reviewer 1 Report

Comments and Suggestions for Authors

Comments on the Quality of English Language

 The English expression of this article should be polished.

Reviewer 2 Report

Comments and Suggestions for Authors

1- The title of Section 3 should be kept with the section text in the same page.

2- The parts of Figure 4 should be kept together with the figure caption in the same page.

3- The superscripts i and i+1 appearing in equation (15) should be defined.

4- The superscripts H appearing in equation (16) should be defined.

5- Define the dataset matrix, I, in equation (26).

6- Do not use the same symbol “I” for two different values the same paper, it is used to designate the iteration number and to designate the dataset matrix in equation (26).

7- Other imaging performance metrics, in addition to the entropy, should be applied to the resulting image.

8 - Comparisons with other recently published results should be included for fair assessment of the proposed method.

Comments on the Quality of English Language

Please, some sentences are very long, try to break them into shorter sentences.

Round 2

Reviewer 2 Report

Comments and Suggestions for Authors

The authors addressed my comments. Thank you.

Author Response

Thanks again for your valuable comment.